# A novel HIV triple broadly neutralizing antibody (bNAb) combination-based passive immunization of infant rhesus macaques achieves durable protective plasma neutralization levels and mediates anti-viral effector functions

Sedem Dankwa[1], Christina Kosman[1], Maria Dennis[1], Elena E. Giorgi[2], Kenneth Vuong[1], Ioanna Pahountis[1], Ashley Garza[3,4], Christian Binuya[1], Janice McCarthy[5,6], Bryan T. Mayer[2], Julia T. Ngo[7], Chiamaka A. Enemuo[7], Diane G. Carnathan[7], Sherry Stanfield-Oakley[3,4], Stella J. Berendam[4], Carolyn Weinbaum[1], Kathleen Engelman[8], Diogo M. Magnani[8], Cliburn Chan[5,6], Guido Ferrari[3,4], Guido Silvestri[7,9], Rama R. Amara[7,10], Ann Chahroudi[11,12], Sallie R. Permar[1,13], Genevieve G. Fouda[1,13], Ria Goswami[1,13]*

1 Department of Pediatrics, Weill Cornell Medicine, New York, NY, United States of America, 2 Vaccine and Infectious Disease Division, Fred Hutchinson Cancer Research Center, Seattle, WA, United States of America, 3 Department of Surgery, Duke University School of Medicine, Durham, NC, United States of America, 4 Duke Human Vaccine Institute, Duke University School of Medicine, Durham, NC, United States of America, 5 Department of Biostatistics and Bioinformatics, Duke University, Durham, NC, United States of America, 6 Center for Human Systems Immunology, Duke University, Durham, NC, United States of America, 7 Division of Microbiology and Immunology, Emory National Primate Research Center, Emory University, Atlanta, GA, United States of America, 8 Department of Medicine, UMass Chan Medical School, Worcester, MA, United States of America, 9 Department of Pathology and Laboratory Medicine, Emory University School of Medicine, Atlanta, GA, United States of America, 10 Department of Microbiology and Immunology, Emory University School of Medicine, Atlanta, GA, United States of America, 11 Department of Pediatrics, Emory University School of Medicine, Atlanta, GA, United States of America, 12 Center for Childhood Infections and Vaccines of Children's Healthcare of Atlanta and Emory University, Atlanta, GA, United States of America, 13 Gale and Ira Drukier Institute for Children's Health, Weill Cornell Medicine, New York, NY, United States of America

☯ These authors contributed equally to this work.
* rig4007@med.cornell.edu

## Abstract

To eliminate vertical HIV transmission and achieve therapy-free viral suppression among children living with HIV, novel strategies beyond antiretroviral therapy (ART) are necessary. Our group previously identified a triple broadly neutralizing antibody (bNAb) combination comprising of 3BNC117, PGDM1400 and PGT151 that mediates robust *in vitro* neutralization and non-neutralizing effector functions against a cross-clade panel of simian human immunodeficiency viruses (SHIVs). In this study, we evaluated the safety, pharmacokinetics, and antiviral potency of this bNAb combination in infant rhesus macaques (RMs). We demonstrate that subcutaneous infusion of the triple bNAb regimen was well tolerated in pediatric monkeys and resulted in durable systemic and mucosal distribution. Plasma obtained from passively-immunized RMs demonstrated potent HIV-neutralizing and Fc-

**Data Availability Statement:** All data files are available from the Github database (https://github.com/Riagoswami1/Triple_bNAb_PK_infant_NHP).

**Funding:** "This work was supported by NIH P01 5P01AI131276 (S.R.P, G.G.F, A.C, G.S, R.R.A. C.C), NIH 5UM1AI164566 (A.C, S.R.P, G.G.F), NIH 5P01AI117915 (S.R.P). and NIH's Office of the Director, Office of Research Infrastructure Programs, P51OD011132. The ADCC assay was performed at Duke Human Vaccine Institute (Durham, NC) as part of the Primate AIDS Vaccine Evaluation Group (PAVEG) contract HSN272201800004C. The rhesusization, expression, and purification of antibodies were partially funded by NIH P40 OD028116 (D.M.M., K.E.). The funders had no role in study design, data collection and analysis, decision to publish, or preparation of the manuscript."

**Competing interests:** "S.R.P serves as a consultant for Merck, Moderna, Pfizer, Hoopika, Dynavax, and GSK on their CMV vaccine programs, and has led sponsored programs with Merck and Moderna on CMV vaccines. Other authors have no conflict of interest to disclose. This does not alter our adherence to PLOS ONE policies on sharing data and materials."

mediated antiviral effector functions. Finally, using the predicted serum neutralization 80% inhibitory dilution titer ($PT_{80}$) biomarker threshold of >200, which was recently identified as a surrogate endpoint for evaluation of the preventative efficacy of bNAbs against mucosal viral acquisition in human clinical trials, we demonstrated that our regimen has $PT_{80}$>200 against a large panel of plasma and breast milk-derived HIV strains and cross-clade SHIV variants. This data will guide the development of combination bNAbs for eliminating vertical HIV transmission and for achieving ART-free viral suppression among children living with HIV.

## Introduction

Antiretroviral therapy (ART) during pregnancy and lactation along with prophylaxis of infants at-risk of HIV exposure has dramatically reduced the burden of vertical HIV transmissions. Despite this success, in 2023, 120,000 infants became infected with the virus globally [1]. Challenges such as late presentation to prenatal care [2], lack of ART adherence [3], and acute maternal infection during pregnancy and/or breastfeeding [4] contribute to the ongoing vertical transmissions of HIV. Hence, ART alone will not be sufficient to eliminate pediatric HIV infections. Furthermore, while ART can efficiently suppress the replicating virus, it cannot clear the HIV reservoir that is established immediately after infection [5]. Thus, virus acquisition in early life forces a child to remain on lifelong uninterrupted ART. Lifelong ART not only poses a huge financial burden [6] and risk of clinical toxicity [7] but maintaining drug adherence is also challenging, especially during adolescence [8]. Therefore, to eliminate pediatric HIV acquisition and to achieve ART-free HIV virologic suppression in children, adjunctive or alternative antiviral strategies will be necessary.

HIV-specific broadly neutralizing antibodies (bNAbs) possess both viral neutralizing properties as well as non-neutralizing effector functions that contribute to their antiviral potency [9], making them attractive therapeutic agents for both HIV prevention and cure. Therefore, several studies have investigated next-generation HIV-specific bNAbs in pre-clinical animal models [10, 11] and human clinical trials [12–14]. For instance, in the recent Antibody-Mediated Prevention (AMP) trials, a CD4-binding site-specific bNAb, VRC01, was demonstrated to protect adults from acquisition of VRC01-sensitive HIV isolates [12], demonstrating the potential of bNAb-based prophylaxis. Several other bNAbs have also been evaluated in clinical trials and shown to have potential preventative or therapeutic benefits [15–17]. A major limitation of the success of single bNAb-based passive immunization strategies has been the pre-existence or emergence of bNAb-resistant HIV variants [12, 13, 15].

To reduce the development of bNAb-resistant HIV variants and potently suppress a highly diverse population of viral variants, bNAb combinations, consisting of antibodies targeting multiple regions of the HIV envelope will be crucial. These regimens may ensure continuous antiviral efficacy by overcoming preexisting resistant variants against a specific bNAb and impeding the development of escape variants against a single bNAb. To date, while combination bNAb regimens have been tested in several adult clinical trials [16, 18], there is only one published pediatric trial, where safety and therapeutic potency of a combination bNAb regimen consisting of VRC01-LS and 10–1074 was demonstrated [19]. Since the Adsorption, Distribution, Metabolism, Elimination and Toxicity (ADMET) profiles of therapeutics can be different in children compared to adults [20], the safety and PK of bNAb combinations in children may be distinct from that of adults. As highly efficacious combination bNAbs are being

developed, assessment of the pharmacokinetics (PK) and pharmacodynamics (PD) of these combinations in the pediatric population is critically needed.

Our group previously demonstrated that a triple bNAb combination consisting of 3BNC117 [21] (CD4 binding site-specific), PGDM1400 [22] (V2 glycan-specific), and PGT151 [23] (gp120-41-specifc) confers robust *in vitro* neutralization breadth and non-neutralizing Fc-mediated effector functions against cross-clade SHIVs [24]. In the current study, we utilized an infant rhesus macaque (RM) model to assess the safety, PK, and *ex vivo* antiviral potency of this triple bNAb combination in the pediatric setting. The pharmacological profile obtained from this study will guide the development of future preclinical and human clinical trials assessing the efficacy of bNAb combinations to prevent vertical HIV transmission or achieve ART-free viral suppression in pediatric populations.

## Results

### Subcutaneous (s.c) infusion of a rhesusized (Rh) triple bNAb formulation in infant RMs

To reduce cross-species immunogenicity of human-derived bNAbs in RMs, the IgG1 heavy and light chains of 3 human isolated bNAbs, 3BNC117 [21], PGDM1400 [22] and PGT151 [23] were engineered to their Rh versions, as described previously [25]. The human (Hu) and Rh versions of the antibodies (Abs) showed comparable *in vitro* neutralization potency, both individually and as a combination (**Fig 1A**). Furthermore, the Rh bNAbs had either similar (3BNC117 and PGDM1400) or more potent (PGT151) antibody dependent cellular cytotoxicity (ADCC) functions compared to the Hu versions (**Fig 1B**), suggesting that rhesusization of the antibodies did not dampen their polyfunctional antiviral potency. To assess the safety and PK of the triple bNAb formulation *in vivo*, we subcutaneously immunized 3 SIV/SHIV-negative infant RMs with 40mg/kg of each of the Rh-bNAbs and followed the animals for 84 days post bNAb infusion (**Fig 1C**). Plasma, saliva and rectal swabs were collected from these animals frequently, following infusion (**Fig 1C**). Animals were monitored closely for adverse outcomes and their body weights were noted regularly. The infant RMs demonstrated a steady gain of body weight over the 76 days of follow-up post-infusion (**Fig 1D**). Additionally, the RMs did not show any physical signs of adverse health outcomes, indicating that s.c administration of the combination bNAbs was safe in this infant model.

### PK of the administered bNAbs in plasma and mucosal compartments and development of anti-drug antibody (ADA)

To estimate the concentrations of the three bNAbs in infant RM plasma after s.c infusion, we performed an anti-idiotype-specific ELISA against each of the infused bNAbs. All 3 infused bNAbs were detected in plasma within 1 hr (0.04 days) after infusion (**Fig 2A–2C**). While the sparse blood sampling early after infusion, due to weight-related blood volume restrictions, precluded the determination of the exact timing to achieve peak antibody concentration, the median time (minimum, maximum) required for 3BNC117, PGDM1400 and PGT151 to reach their maximum observed plasma concentration were 1 day (0.25, 3), 1 day (1,1) and 2 days (1,3), respectively (**Table 1** and **Fig 2A–2C**). Interestingly, despite administration of equivalent doses of the 3 bNAbs in the combined formulation (40mg/kg per bNAb), the maximum observed plasma concentration ($C_{Max}$) of the three bNAbs varied. 3BNC117 achieved a higher median $C_{Max}$ (minimum, maximum) of 652.85 µg/ML (573.51, 735.61), followed by PGDM1400 of 328.6 µg/ML (321.4, 512.94) and PGT151 of 255.73 µg/ML (123.49, 260.51). The median elimination half-life ($t_{1/2}$) of the 3 bNAbs were comparable to each other, with the

### A. SHIV.C.CH505.375H.dCT neutralization

| Neutralization titer IC$_{80}$ (µg/ML) | | | |
|---|---|---|---|
| Specificity | BNAbs | Human (Hu) | Rhesusized (Rh) |
| CD4-binding site | 3BNC117 | 0.21 | 0.17 |
| Gp120-gp41 interface | PGT151 | 0.21 | 0.34 |
| V2-glycan | PGDM1400 | 0.01 | 0.03 |
| Multiple | 3BCN117+PGT151+PGDM1400 | 0.02 | 0.04 |

| IC$_{80}$ (µg/ML) | |
|---|---|
| | ≤0.04 |
| | >0.04 |

### B. HIV CH505 T/F ADCC

| ADCC antibody titer (µg/ML) | | | |
|---|---|---|---|
| Specificity | BNAbs | Human (Hu) | Rhesusized (Rh) |
| CD4-binding site | 3BNC117 | 0.15 | 0.27 |
| Gp120-gp41 interface | PGT151 | 2.44 | 0.32 |
| V2-glycan | PGDM1400 | >50 | >50 |

| ADCC titer (µg/ML) | |
|---|---|
| | <1 |
| | 1-50 |
| | >50 |

### C.

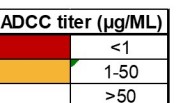

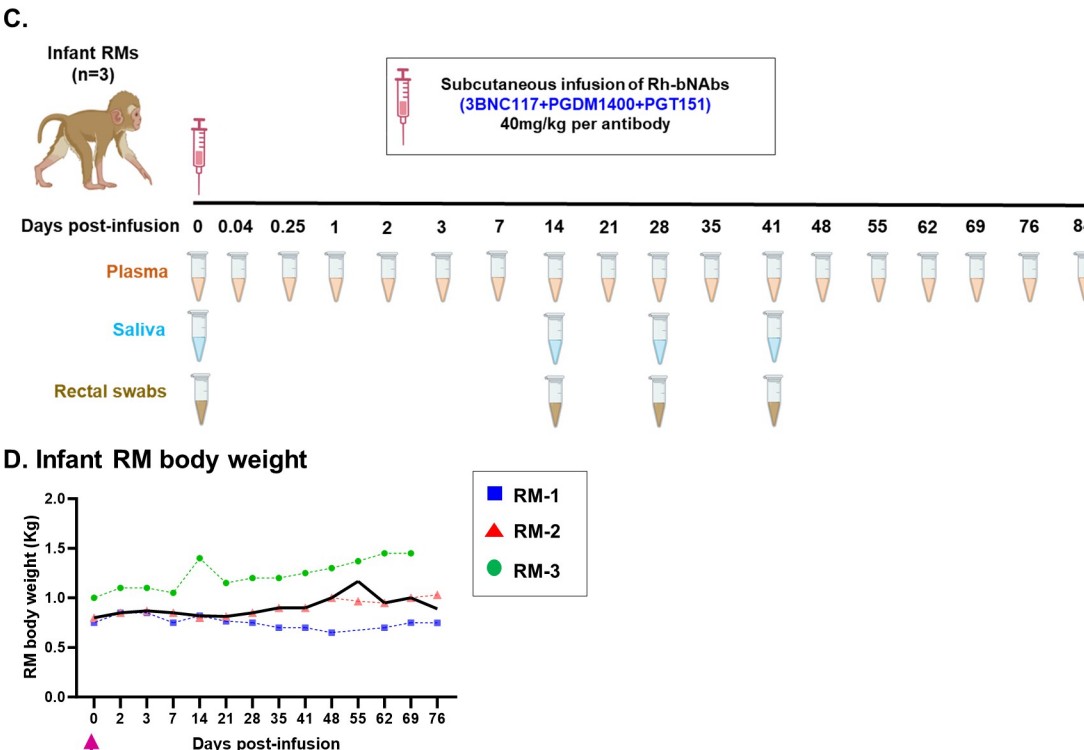

### D. Infant RM body weight

**Fig 1. Passive immunization of a rhesusized (Rh)-triple bNAb combination in infant rhesus macaques (RM). (A)** Neutralization of SHIV.C.CH505.375H.dCT by the human (Hu) and rhesusized (Rh) versions of 3 bNAbs, 3BNC117, PGDM1400, PGT151 and their combination *in vitro*. Neutralizing IC$_{80}$s (µg/mL) of the antibodies are reported. **(B)** HIV CH505 T/F-specific ADCC response of the Hu and Rh versions of the 3 above-mentioned bNAbs. ADCC antibody titers in µg/mL are reported. **(C)** Three infant RMs passively immunized with a triple bNAb formulation consisting of Rh-3BNC117,—PGDM1400 and -PGT151, each antibody at a 40mg/kg dosage. Plasma, saliva, and rectal swab samples collected at indicated time points for experimental analysis. **(D)** Body weights of the infant RMs monitored over 76 days post-infusion. Each RM is indicated by a symbol and the median body weight over time is represented as bolded line. Pink arrow indicates the time of triple Rh-bNAb infusion.

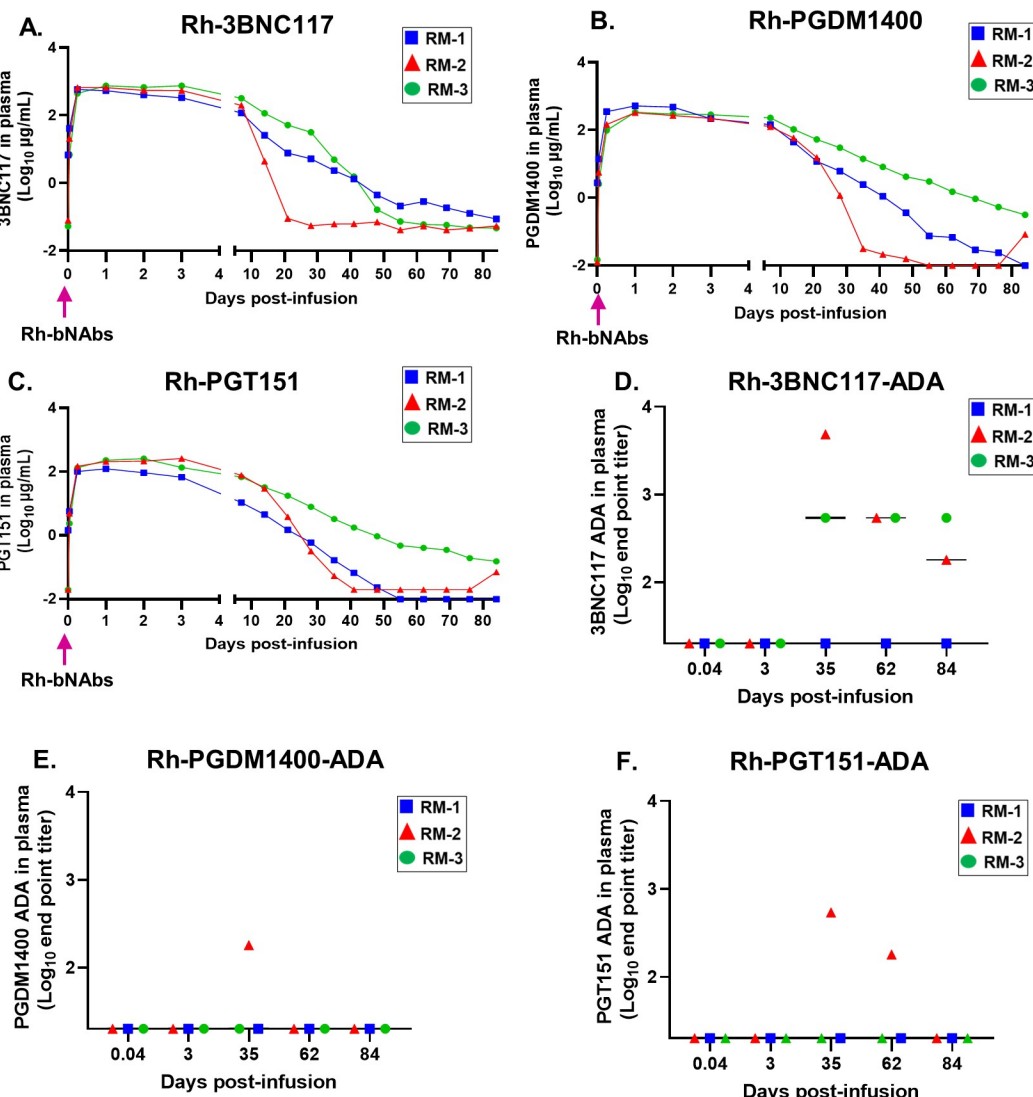

**Fig 2. Concentrations of infused Rh-bNAbs in plasma and development of anti-drug antibody (ADA) in infant RMs.**
Plasma concentration of passively infused (**A**) Rh-3BNC117, (**B**) Rh-PGDM1400 and (**C**) Rh-PGT151 at indicated time points post-infusion, as determined by ELISA. ADA responses against (**D**) Rh-3BNC117, (**E**) Rh-PGDM1400, and (**F**) Rh-PGT151 at indicated time points, as measured by ELISA. Each monkey is denoted by a unique symbol. Pink arrow indicates the time of triple Rh-bNAb infusion.

median $t_{1/2}$ for 3BNC117, PGDM1400 and PGT151 being 6.51 days, 5.47 days and 7.05 days, respectively (**Table 1**). Additional PK parameters such as area under the curve (AUC), mean residence time (MRT) and clearance (CL) of the bNAbs in plasma were computed using the plasma antibody (Ab) ELISA data and are outlined in **Table 1**. Finally, using a semi-quantitative assay, as described in the methods, we demonstrated detectable levels of the 3 bNAbs in the saliva and rectal swabs of the infant monkeys by day 14 post-infusion, which gradually waned over time and became mostly undetectable by 41 days post-infusion (**Table 2**). These data suggest that after s.c. infusion, the bNAbs penetrated to the mucosal sites of HIV exposure and might provide local protection against mucosal viral acquisition for >14 days.

Surprisingly, one animal (RM-2) demonstrated a steeper decline of all 3 bNAb levels in plasma, compared to the other 2 animals (RM-1 and RM-3). We, therefore, sought to test if

**Table 1. Pharmacokinetic measures of subcutaneously administered Rh-3BNC117, Rh- PGDM1400 and Rh-PGT151 in plasma of infant RMs.**

| Pharmacokinetic parameters | 3BNC117 | | | PGDM1400 | | | PGT151 | | |
|---|---|---|---|---|---|---|---|---|---|
| | RM-1 | RM-2 | RM-3 | RM-1 | RM-2 | RM-3 | RM-1 | RM-2 | RM-3 |
| Dose (mg/kg) | 40 | 40 | 40 | 40 | 40 | 40 | 40 | 40 | 40 |
| Animal weight (kg) | 0.80 | 0.85 | 1.00 | 0.80 | 0.85 | 1.00 | 0.80 | 0.85 | 1.00 |
| CL (mL.day/kg) | 0.011 | 0.009 | 0.006 | 0.011 | 0.014 | 0.010 | 0.061 | 0.019 | 0.024 |
| MRT (day) | 4.95 | 3.58 | 7.10 | 5.89 | 6.53 | 11.03 | 3.97 | 5.15 | 8.28 |
| $AUC_{0-84}$ (µg/ML.day) | 2899.00 | 3869.66 | 6498.43 | 2869.27 | 2374.08 | 4084.51 | 523.90 | 1780.12 | 1684.10 |
| $C_{Max}$ (µg/ML) | 573.51 | 652.85 | 735.61 | 512.94 | 321.40 | 328.60 | 123.49 | 260.51 | 255.73 |
| $T_{Max}$ (day) | 0.25 | 1 | 3 | 1 | 1 | 1 | 1 | 3 | 2 |
| $T_{1/2}$ (day) | 12.71 | 6.51 | 5.32 | 5.47 | 5.17 | 8.90 | 5.96 | 7.05 | 8.20 |

Abbreviations: CL, Clearance; MRT, Mean residence time; $AUC_{0-84}$, Area under the concentration-time graph from day 0 to day 84; $C_{Max}$, Maximum observed concentration; $T_{Max}$, Time when the maximum concentration was observed; $T_{1/2}$, Elimination half-life.

this steeper decline was due to a concomitant development of ADA responses against the infused bNAbs. As expected, RM-2 demonstrated robust ADA response against all 3 bNAbs at 35 days post-infusion (Fig 2D–2F), the time when the plasma concentrations of the 3 bNAbs declined to almost undetectable levels in this animal (Fig 2A–2C). Similarly, in RM-3, development of ADA response against 3BNC117 was associated with decline in the concentration of 3BNC117 in plasma. Interestingly, RM-1 did not develop ADA response against any of the infused bNAbs, suggesting that host-specific factors might contribute to the development of ADAs. The developed ADAs in the macaques were found to be specific for the F(ab)2 region, but not the Fc region of each of the infused bNAbs (S1 Fig).

**Table 2. Detection of Rh-bNAbs in the mucosal compartments of passively infused infant RMs.**

| BNAb | Mucosal secretion | Days post-infusion | bNAb-specific IgG (ng/mL) | | |
|---|---|---|---|---|---|
| | | | RM-1 | RM-2 | RM-3 |
| Rh-3BNC117 | Saliva | 14 | ++ | + | +++ |
| | | 28 | + | - | + |
| | | 41 | - | - | UA |
| | Rectal swab | 14 | ++ | + | ++ |
| | | 28 | ++ | + | - |
| | | 41 | - | + | - |
| Rh-PGT151 | Saliva | 14 | +++ | ++ | +++ |
| | | 28 | + | - | + |
| | | 41 | - | - | - |
| | Rectal swab | 14 | + | ++ | ++ |
| | | 28 | ++ | + | - |
| | | 41 | - | + | - |
| Rh-PGDM1400 | Saliva | 14 | +++ | +++ | +++ |
| | | 28 | ++ | + | ++ |
| | | 41 | - | - | ++ |
| | Rectal swab | 14 | - | +++ | ++ |
| | | 28 | +++ | + | - |
| | | 41 | - | - | - |

Abbreviations: UA indicates sample unavailable; -: bNAb levels below detection limit of the assay (24.41ng/mL); +: bNAb levels 24.41-100ng/mL; ++: bNAb levels 100-500ng/mL; +++: bNAb levels >500ng/mL.

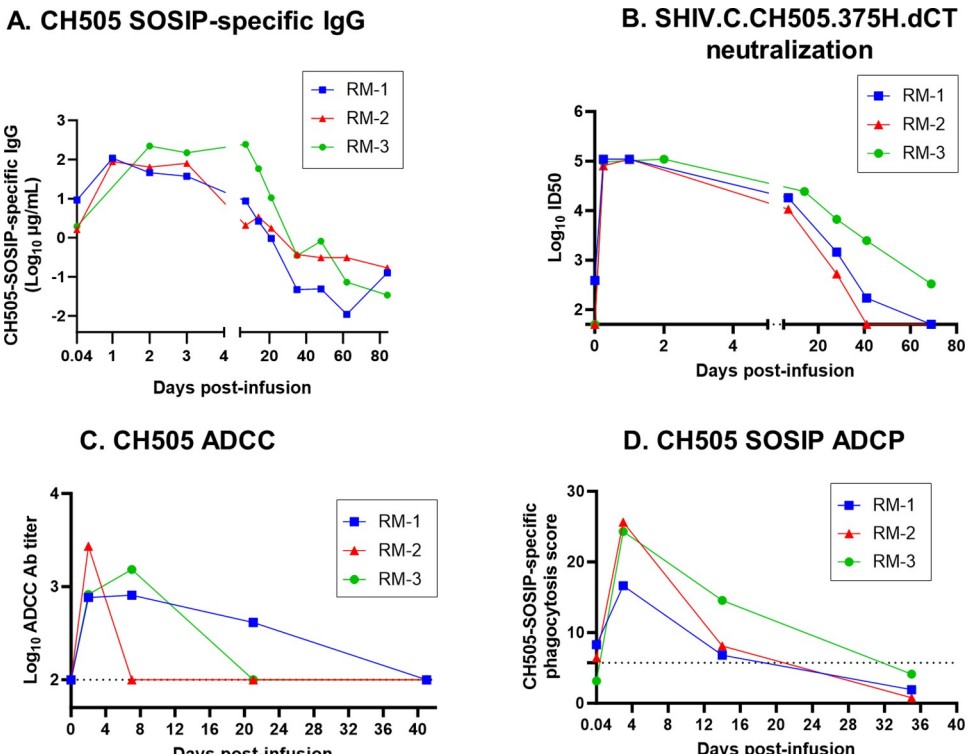

**Fig 3. Specificity, neutralizing and non-neutralizing antibody functions of plasma from passively-infused infant RMs. (A)** Magnitude and kinetics of HIV CH505-SOSIP-specific IgG antibodies in the plasma of triple Rh bNAb-infused infant RMs. **(B)** Plasma neutralization titer against a tier-2 clade C virus, SHIV.C.CH505.375H.dCT, at indicated time points. **(C)** Antibody-dependent cellular cytotoxicity (ADCC) Ab titers against HIV CH505 T/F infectious molecular clone-infected target cells and **(D)** HIV CH505 SOSIP-specific antibody-dependent cellular phagocytosis (ADCP) scores of plasma from combination Rh-bNAb-infused infant RMs. Each monkey is denoted by a color and a unique symbol. The dashed lines indicate the limit of detection (LOD) of ADCC and ADCP assays. LOD for ADCC assay is 100 and LOD for ADCP assay is defined as 3 times ADCP score of a non-HIV-specific antibody CH65 against HIV CH505 SOSIP.

## HIV envelope (Env) trimer specificity and SHIV neutralization potency of bNAb-infused infant RM plasma

To assess the ability of the infused bNAbs to bind to HIV Env trimers, we performed an HIV CH505 T/F Env trimer (SOSIP)-binding IgG ELISA. Plasma IgG from all 3 infants demonstrated specificity against the native-like HIV CH505 SOSIP Env antigen (**Fig 3A**). Maximum binding to HIV Env trimer was observed within 1–2 days post bNAb-infusion, which was also the time when the bNAbs were present at their maximum levels in the plasma (**Fig 2A–2C**). The HIV Env trimer-specific IgG levels followed a kinetics similar to that of the plasma bNAb levels and gradually declined with the clearance of the infused antibodies from plasma. We next evaluated the ability of the bNAb-infused plasma to neutralize a tier-2 clade C next-generation SHIV (SHIV.C.CH505.375H.dCT), that can infect and replicate in infant RMs [26] as well as recapitulate the viral replication dynamics and immunopathogenesis of HIV infection in humans [27]. Plasma from all 3 RMs potently neutralized SHIV.C.CH505.375H.dCT *in vitro*, with a maximum neutralization potency (50% inhibitory dose, $ID_{50}$) between 6hrs (0.25 days) and 2 days post-infusion ($ID_{50}$ range: 80363.4–109350) (**Fig 3B**). RM-2, the animal that developed ADA against all 3 bNAbs (**Fig 2D–2F**), exhibited a steeper decline in HIV neutralization $ID_{50}$ titers, compared to other 2 animals, due to waning of most of the plasma bNAb

levels. Interestingly, plasma from RM-3 demonstrated most durable neutralization response, with detectable $ID_{50}$ titer even at 69 days post-infusion, a time point when 2/3 bNAbs (PGT151 and PGDM1400) were still detectable in the plasma of that animal (**Fig 2A–2C**).

## Non-neutralizing antibody-mediated effector functions of the passively immunized infant RM plasma

Our group previously demonstrated that a combination of 3BNC117, PGDM1400 and PGT151 can mediate potent *in vitro* antibody-dependent cellular cytotoxicity (ADCC) and antibody-dependent cellular phagocytosis (ADCP) effector functions against cross clade SHIVs [24]. Here, we assessed the non-neutralizing effector functions of the plasma obtained from triple bNAb-infused infant RMs. First, we evaluated the ADCC activity of the plasma using HIV CH505 T/F-infected human lymphoblast (CEM.NKR.CCR5+) cells as targets and human PBMCs as effectors. Maximum ADCC Ab titers in plasma were detected by day 2 (RM-2) or day 7 (RM-1 and RM-3) which steadily declined over time with clearance of the bNAbs from plasma (**Fig 3C**). Not surprisingly, in RM-2, the animal that developed ADA against all 3 infused bNAbs, plasma ADCC titer reached undetectable levels faster, compared to RM-1 or RM-3.

We also measured the ability of the infused plasma to mediate ADCP against beads coated with HIV CH505 T/F SOSIP. Maximum ADCP scores were observed at day 3 post-infusion for all 3 animals that reached a median score (minimum, maximum) of 24.3% (16.6%, 25.6%) (**Fig 3D**). Similar to neutralizing and ADCC responses, ACDP response in plasma also waned as plasma bNAb levels declined.

## Predicted serum neutralization 80% inhibitory dilution titer ($PT_{80}$) and neutralization coverage of the infused triple bNAb regimen against cross-clade SHIV and HIV variants

Using the AMP trial participant data [12], $PT_{80}$ was recently identified as a biomarker that can predict the preventative efficacy of a bNAb regimen against mucosal HIV acquisition [28], and was found to be correlated with the experimental plasma $ID_{80}$. Here, we interrogated if this established relationship between plasma $ID_{80}$ and $PT_{80}$ of the infused triple bNAbs is also observed in the pediatric setting. To this end, we estimated the $PT_{80}$ of the triple bNAb combination against SHIV.C.CH505.375H.dCT at several time points post-infusion, as described in the methods, and correlated the values with the experimental plasma $ID_{80}$, at corresponding timepoints. As observed in the adult human clinical trial [28], the experimental plasma $ID_{80}$ titers against SHIV.C.CH505.375H.dCT was well correlated with the $PT_{80}$ of the infused triple bNAb combination (Kendell's Tau = 0.89; p<0.001) (**Fig 4A**), confirming the use of $PT_{80}$ biomarker as a correlate of neutralization potency ($ID_{80}$) of a bNAb regimen in infants.

The AMP trial data suggested that if the $PT_{80}$ of a bNAb regimen to an exposed HIV is >200, the preventative efficacy to block vaginal/rectal acquisition of that virus will be ≥90% [28]. Although a similar study has not yet been done in the setting of breast milk transmission in infants, we applied this published human clinical trial data as a benchmark to estimate the neutralization efficacy of our triple bNAb combination. We assessed the duration for which the $PT_{80}$ of the infused triple bNAb regimen remained above 200 levels in plasma, against a panel of 9 cross-clade tier-2 SHIV variants. The $PT_{80}$ of the combination bNAb regimen remained >200 for a longer timeframe for SHIVs that could be potently neutralized by the bNAb combination, compared to those that were resistant to neutralization (**Fig 4B and S2A– S2I Fig**). Importantly, a $PT_{80}$>200 was achieved for 7/9 (77.8%) of the tested SHIV variants, highlighting the efficacy of the tested triple bNAb regimen in neutralizing cross-clade SHIVs.

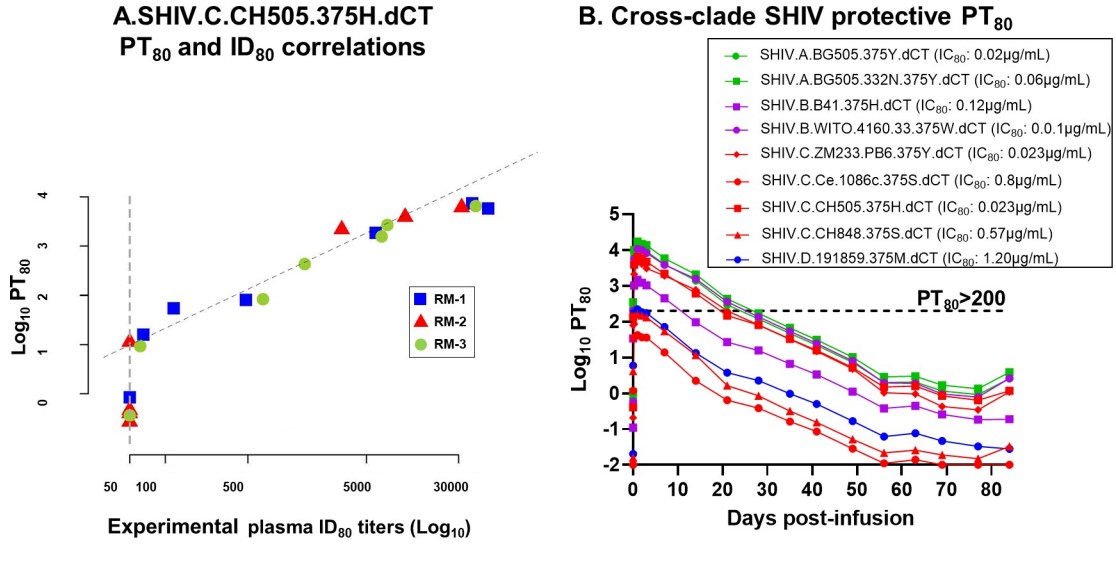

**A. SHIV.C.CH505.375H.dCT PT$_{80}$ and ID$_{80}$ correlations**

**B. Cross-clade SHIV protective PT$_{80}$**

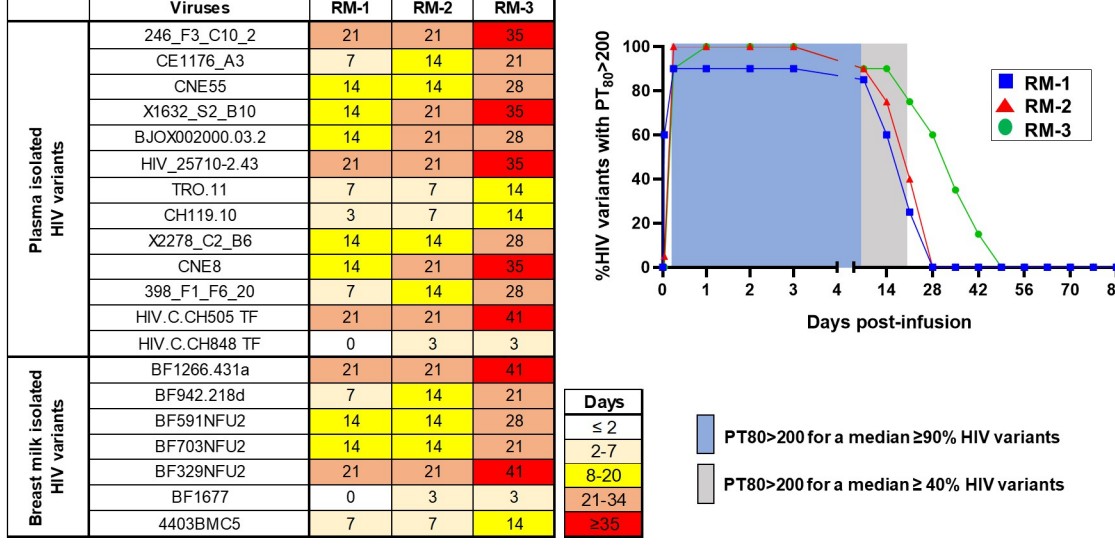

**C. Days for which PT80>200 for HIV**

| | Viruses | RM-1 | RM-2 | RM-3 |
|---|---|---|---|---|
| Plasma isolated HIV variants | 246_F3_C10_2 | 21 | 21 | 35 |
| | CE1176_A3 | 7 | 14 | 21 |
| | CNE55 | 14 | 14 | 28 |
| | X1632_S2_B10 | 14 | 21 | 35 |
| | BJOX002000.03.2 | 14 | 21 | 28 |
| | HIV_25710-2.43 | 21 | 21 | 35 |
| | TRO.11 | 7 | 7 | 14 |
| | CH119.10 | 3 | 7 | 14 |
| | X2278_C2_B6 | 14 | 14 | 28 |
| | CNE8 | 14 | 21 | 35 |
| | 398_F1_F6_20 | 7 | 14 | 28 |
| | HIV.C.CH505 TF | 21 | 21 | 41 |
| | HIV.C.CH848 TF | 0 | 3 | 3 |
| Breast milk isolated HIV variants | BF1266.431a | 21 | 21 | 41 |
| | BF942.218d | 7 | 14 | 21 |
| | BF591NFU2 | 14 | 14 | 28 |
| | BF703NFU2 | 14 | 14 | 21 |
| | BF329NFU2 | 21 | 21 | 41 |
| | BF1677 | 0 | 3 | 3 |
| | 4403BMC5 | 7 | 7 | 14 |

| Days |
|---|
| ≤ 2 |
| 2-7 |
| 8-20 |
| 21-34 |
| ≥35 |

**D. PT$_{80}$>200 against HIV variants**

PT80>200 for a median ≥90% HIV variants

PT80>200 for a median ≥ 40% HIV variants

**Fig 4. PT$_{80}$ of the infused bNAb regimen against cross-clade SHIV and HIV variants. (A)** Correlation of the triple bNAb combination PT$_{80}$ and experimental plasma ID$_{80}$ against SHIV.C.CH505.375H.dCT for each RM over multiple time points. Each animal is represented by a single color and symbol. Kendal Tau's correlation was performed to calculate the correlation coefficient and p-value. **(B)** Median PT$_{80}$ of the triple bNAb combination against 9 cross-clade SHIV variants, consisting of clade A, B, C and D pseudoviruses. Dashed line indicates PT$_{80}$>200. Neutralization IC$_{80}$ of the triple bNAb regimen against corresponding SHIVs are included in the figure key. **(C)** Heat map representing the number of days for which PT$_{80}$ of the triple bNAb combination remains above 200 for plasma and breast milk isolated HIV variants. **(D)** Percentage of HIV variants for which PT80>200 for 3 RMs at indicated time points. Each animal is denoted by a unique symbol. Shaded boxes indicate the range of days post-infusion for which PT$_{80}$>200 against a median of ≥90% (blue) and ≥40% (light grey) HIV variants are achieved.

RM-3, the animal that demonstrated most durable plasma neutralization titer (ID$_{80}$) (**Fig 3B**) maintained a PT$_{80}$>200 for longer duration compared to the other two animals, further validating the positive correlation of PT$_{80}$ biomarker with ID$_{80}$ levels.

While our study was performed in NHPs, and did not include a viral challenge group to monitor the efficacy of the passive immunization regimen in achieving viral protection, we

used the published human $PT_{80}$ benchmark (>200) to compute the potential neutralization efficacy of the infused triple bNAb combination against a panel of 20 pseudoviruses, including 11 human plasma-isolated globally circulating variants [29], 2 T/F HIV variants and 7 human breast milk-isolated HIV variants [30]. We first computed the duration of time for which the triple bNAb combination maintained a $PT_{80}$>200 post-infusion, against these HIV variants. Our analysis indicated that $PT_{80}$ of the triple bNAb regimen remained >200 for a median of 14 days for both the plasma-isolated and breast milk-isolated variants (**Fig 4C**). Interestingly, RM-3, the animal with the most durable neutralization titer (**Fig 3B**), exhibited $PT_{80}$>200 for an extended period (plasma and breast milk virus median: 28 days), compared to other 2 animals (median: 14 days). Finally, we estimated the potential neutralization efficacy of the bNAb regimen by evaluating the number of plasma- and breast milk-isolated HIV variants for which its $PT_{80}$ remained above 200, post-infusion. The combination bNAb regimen had a $PT_{80>}200$ for ≥90% and ≥40% HIV variants for a median of 7 days and 21 days, respectively (**Fig 4D**). As expected, for RM-3, the bNAb regimen had a $PT_{80}$>200 for ≥90% HIV variants for an extended time (14 days), compared to RM-1 (3 days) and RM-2 (7 days).

## Discussion

To eliminate vertical transmission of HIV during breastfeeding and to achieve therapy-free long-term viral control in children living with HIV, ART alone will not be sufficient. Passive immunization of infants and children with bNAbs is currently being investigated as a potential adjunct or alternative to ART [19, 31]. In fact, recently, using computational modeling on HIV-exposed infants, it was projected that bNAb-based prophylaxis would reduce the cumulative pediatric HIV incidence in Sub Saharan African regions and would be cost-effective when added to the current prevention strategies [32]. By systematic assessment of the neutralizing and non-neutralizing antiviral functions of bNAbs targeting multiple epitopes of the HIV Env, we previously identified a bNAb combination, 3BNC117 [21], PGDM1400 [22], and PGT151 [23] to have potent *in vitro* polyfunctional antiviral potency and breadth against cross-clade SHIVs [24], making this combination ideal for studies in the infant RM model. In the current study, we passively immunized infant RMs with this triple bNAb regimen and monitored the safety and PK of the antibodies in an infant setting. We found that the passively infused bNAbs were well tolerated in these infant monkeys and exhibited robust systemic and mucosal antibody levels. Additionally, plasma from passively immunized monkeys showed neutralization breadth against systemically circulating and breast milk-isolated HIV strains, as well as potent non-neutralizing functions, highlighting the potential efficacy of the triple bNAb regimen in suppressing mucosally-transmitted HIV variants via perinatal or breastfeeding transmission.

As passive immunization with combination bNAbs seems to be a promising approach to suppress HIV replication and achieve viral remission, several adult preclinical and clinical trials are being conducted to assess the safety, PK and efficacy of newly developed combination bNAb regimens [16, 18, 33, 34]. However, to date, there is only one published pediatric clinical trial of a combination bNAb therapeutic regimen [19] and a handful published or ongoing trials using single bNAbs as prophylactics [31, 35, 36]. The most common approach for estimating the dose and PK of any new therapeutic regimen for a pediatric trial has been based on extrapolation of available adult data. Since the physiology and anatomy of children are quite different from adults, differences in metabolic and drug absorption rates in children can affect the pharmacokinetics, antiviral functions and safety of bNAb regimen differently in children, compared to adults [37]. In fact, in a passive immunization trial of long-acting VRC01 in infants, a faster absorption of antibodies after s.c. infusion and reduced half-life of the biologics

was observed compared to an adult trial [31]. Hence, as new potent combination bNAb regimens are developed, assessing their safety and PK distribution in the pediatric setting will be crucial. While it is logistically challenging to conduct pediatric human clinical trials, SHIV challenge infant RM models, that can mimic the HIV immunopathogenesis in humans [13], can be an instrumental preclinical model for assessing safety, dosing and efficacy to inform the design of human infant clinical trials.

While we are the first group to include PGT151 in a combination bNAb regimen, the safety, PK and efficacy of administering 3BNC117 and PGDM1400 has been previously reported [16, 18, 34, 38]. To improve the logistical challenges of i.v. immunization in the pediatric population, we have administered the bNAbs subcutaneously, similar to a previous infant bNAb prophylaxis trials [31, 36]. Similar to other groups, we showed that both 3BNC117 and PGDM1400 were well tolerated and resulted in detectable systemic distribution [15, 16]. Interestingly, while 3 bNAbs were administered at the same dosage (40mg/kg), the achieved plasma levels for the 3 Abs were not similar, with 3BNC117 having a higher $C_{max}$, compared to the other 2 Abs, perhaps due to differences in rates of translocation to the systemic circulation post s.c-infusion. (**Fig 2A–2C**). A similar observation was made in an adult human trial with 3BNC117 and 10–1074 [38], where infusion of the same dose of bNAbs (10mg/kg) resulted in variable systemic Ab levels, suggesting differential absorption and localization of each bNAb in the combination regimen. Comparison of our PK data in infants RMs with previously published adult RM data and human clinical trial data revealed important similarities and dissimilarities. While the systemic bNAb levels in infant monkeys after a single dose Ab infusion were comparable with adult monkey studies [34] and human trials [15], the $t_{1/2}$ of both 3BNC117 [15] and PGDM1400 [16] in our RM cohort was shorter than that observed in HIV naïve humans. Notably, similar to our infant RM data, shorter $t_{1/2}$ of bNAbs was also noted in adult RMs [34], suggesting that the physiology and metabolism of RMs could be contributing to this faster elimination of bNAbs.

In this study, s.c immunization with a single dose of Rh-3BNC117, Rh-PGDM1400, and Rh-PGT151 at a concentration of 40mg/kg per Ab, resulted in development of ADA responses against 3BNC117, PDGM1400 and PGT151, in 2/3 (66.7%), 1/3 (33.3%), and 1/3 (33.3%) animals, respectively (**Fig 2D–2F**). Interestingly, RM-2 developed ADA response against all 3 Abs and RM-3 developed response against only 3BNC117. To date, adult studies with an identical passive immunization regimen have not been performed. Therefore, the contribution of age to the development of ADA response is not clearly understood. In a clinical trial with HIV-uninfected adults, intravenous (i.v) infusion of 10mg/kg 3BNC117 at a single dose resulted in the development of ADA response in 2/6 (33.3%) individuals [38]. While the development of 3BNC117-specific ADA response in our infant RM cohort at a higher frequency than in human adults might be due to age, the contribution of other parameters such as immunization regimen, dose, frequency and route of administration, and host-specificity could not be ignored. For instance, multi-dose s.c infusion of PGT121 and VRC07-523 in SHIV SF162P3-infected infant RMs at 5mg/kg per bNAb resulted in the development of PGT121-specific ADA response in 6/6 (100%) and VRC07-523-specific ADA response in 6/6 (100%) monkeys [33]. In contrast, multi-dose i.v infusion of 30mg/kg of VRC01LS and 10–1074 in children living with HIV did not result in the development of any ADA response against the infused antibodies [19]. These data suggest that ADA response potentially develops at a higher frequency in pediatric RMs than in humans, although immunization regimen, dose, and route of infusion might be regulatory parameters as well.

To protect against HIV acquisition during breastfeeding, presence of passively immunized bNAbs in the mucosal secretions such as infant saliva is crucial. However, over the years, very few studies have investigated the ability of passively immunized bNAbs to localize in mucosal

compartments [39, 40], due to translocation of low levels of Abs that cannot be easily detected using standard assays. To alleviate the issue of low levels of detection in the mucosal compartments, bNAbs have been engineered to enhance neonatal Fc receptor (FcRn) binding that can improve such mucosal localizations [41]. While our study did not use engineered bNAbs, the subcutaneously infused bNAbs localized to mucosal compartments such as rectal swabs and saliva, in 3/3 monkeys, highlighting the penetration of this bNAb regimen to the mucosal sites of HIV exposure, even with s.c. administration.

A combination bNAb regimen, consisting of antibodies with both anti-viral neutralization and non-neutralizing properties will be required for fully suppressing viral replication and clearing tissue reservoir. This strategy will reduce the chances of therapy failure due to development of viral resistance, as each antibody will likely target different epitopes on the HIV Env region to induce neutralizing and non-neutralizing functions. In fact, recently, by screening a large panel of HIV variants, it was suggested that viruses rarely develop contemporary resistance to both neutralizing and non-neutralizing antibody functions [42]. In our study, 3/3 and 2/3 bNAbs demonstrated potent neutralizing properties and ADCC functions *in vitro*, respectively. More importantly, in accordance with data obtained from previous passive immunization trials [16, 38], plasma from triple bNAb-infused infant monkeys from this study also demonstrated potent HIV neutralizing and non-neutralizing antibody functions. In the AMP trial, a neutralization titer biomarker, $PT_{80}$, was identified as a correlate of prevention efficacy of VRC01 and was associated with plasma $ID_{80}$ levels [28]. We, for the first time investigated this correlation between $PT_{80}$ and $ID_{80}$ in a pediatric setting. Furthermore, since availability of plasma in pediatric studies are generally limited to assess neutralization breadth ($ID_{80}$) against multiple cross-clade HIV variants, the $PT_{80}$ biomarker can be an instrumental tool to characterize the breadth and potency of the infused bNAb regimen against globally circulating HIV variants [29] and viruses that are postnatally transmitted via breastfeeding [30]. Similar to previous passive immunization human trials [38], our data also revealed differential potency and durability of neutralization response against tested HIV variants. Importantly, utilizing the $PT_{80}$ biomarker threshold of >200, we demonstrated that our triple bNAb regimen could neutralize plasma and breast milk-isolated HIV variants. As pediatric clinical trials are currently being planned to estimate the efficacy of passive immunization of bNAbs as HIV prophylactics, interrogating whether a $PT_{80}$ biomarker threshold of >200 against perinatally and breast milk-exposed HIV variants is sufficient to provide ≥90% preventative efficacy against vertical HIV transmission will be crucial. Furthermore, our triple bNAb combination regimen demonstrated a PT80>200 for tested HIV variants for at least 7 days, suggesting that multiple infusion of this regimen or infusion with engineered bNAbs with leucine-serine (LS) substitutions that can extend the half-lives of the bNAbs and improve mucosal Ab levels [41, 43], would be useful to prevent HIV transmission during the breastfeeding period and achieve long-term HIV suppression in children living with HIV.

Our study had limitations, such as a very small sample size of 3 monkeys which served as a pilot study for planning a SHIV challenge study that could estimate the efficacy of the novel bNAb combination in preventing or clearing HIV and characterizing viral escape from the infused regimen. Additionally, due to limited sampling capability in the infant monkeys, we could not estimate the distribution of bNAbs in tissues, such as lymph nodes and gut, which are crucial sites of HIV reservoir in infants [44], and would require detectable levels of the bNAbs in these tissues to achieve viral clearance.

In summary, in this study we evaluated the safety, PK and antiviral efficacy of a novel triple bNAb combination, 3BNC117 (CD4 binding site specific), PGDM1400 (V2 glycan specific) and PGT151 (gp120-gp41 interface specific), using a pediatric non-human primate model. In addition to assessing the *in vitro* neutralizing and non-neutralizing effector functions of the

regimen against a single SHIV variant, we estimated the breadth and durability of neutralization response, using a recently established neutralization titer biomarker of efficacy, $PT_{80}$. These findings will guide the optimal designing of a passive immunization regimen to explore their potential as a prophylactic to prevent vertical HIV acquisition or as a therapeutic to achieve long-term HIV control in pediatric population.

## Materials and methods

### Ethics statement and animal care

Rhesus macaque (*Macaca mulatta*) studies were performed at the Emory National Primate Research Centre (ENPRC), in Atlanta, GA, USA, in compliance with all ethical regulations for research. ENPRC animal facilities are accredited by the U.S. Department of Agriculture (USDA) and the Association for Assessment and Accreditation of Laboratory Animal Care (AAALAC) International. Animal care was performed in compliance with the 2011 Guide for the Care and Use of Laboratory Animals provided by the Institute for Laboratory Animal Research. The study protocol and all procedures were approved by the Emory University Institutional Animal Care and Use Committee (IACUC) and animal care facilities. The animals were born at the ENPRC to dams housed in indoor/outdoor group housing. The infants were removed from the dams when they were approximately 2 weeks old and transferred to a nursery, where they were housed in social groups or pair housed for the duration of the study. The infants were fed in accordance with the ENPRC standard operating procedures (SOPs) for nonhuman primate (NHP) feeding. After being removed from the dams, infants were fed ENPRC-approved milk replacer (Similac Advance, OptiGro infant formula with iron, and/or Similac Soy Isomil OptiGro infant formula with iron; Abbott Nutrition, Columbus, OH) until 14 weeks of age. Infants are provided softened standard primate jumbo chow biscuits (jumbo monkey diet 5037; Purina Mills, St. Louis, MO) and a portion of fruit starting around 4 weeks of aged daily. As animals aged, additional enrichment was provided daily, in the form of various fresh produce. Approved procedures ensured that potential distress, pain, discomfort and/or injury were limited to only those unavoidable in the conduct of the research plan. The sedative Ketamine (10 mg/kg) and/or Telazol (4 mg/kg) were administered as necessary for blood and tissue collections with analgesics used when determined appropriate by veterinary medical staff. For the entire duration of the study, animals were checked at least twice a day by animal care technicians and at least daily by the clinical veterinary staff. Physical examinations were performed each time an animal was anesthetized for blood collection or other procedures. For the entire duration of the study, the animals were monitored by regular evaluation of body weight, appetite, and behavior. Animals were euthanized via barbiturate overdose (100 mg/kg pentobarbital) under anesthetic via IACUC guidelines using standard methods consistent with the recommendations of the American Veterinary Medical Association (AVMA) Guidelines for Euthanasia.

### Passive immunization of RMs

Three healthy, SIV-uninfected, Indian infant RMs (1 female and 2 male) of a median (maximum, minimum) age of 69 days (65, 73) were enrolled for the study. Animals were subcutaneously immunized with a single dose of Rh 3BNC117 IgG, PGT151 IgG and PGDM1400 IgG (UMass Biologics, Boston, MA) at 40mg/kg per antibody. Blood was drawn from the RMs, and saliva and rectal swabs were collected using Weck cell surgical spears at time points indicated in **Fig 1C**. Plasma was separated from blood cells, by spinning blood tubes at 2000 rpm for 15 min at 4 ˚C. Samples were stored at -80˚C until further use.

## Pharmacokinetics of Rh bNAbs in plasma

The plasma levels of the passively-infused Rh bNAbs were detected using ELISA. To this end, 384-well polystyrene plates (Corning, NY) were coated overnight with 45ng/well of an anti-idiotypic antibody against Rh-3BNC117, -PGT151 or -PGDM1400 (manufactured by Innovagen, Sweden) in KPL coating solution (SeraCare Life Sciences, Milford MA), washed with wash buffer (1%tween 20 in 1X PBS) and blocked with superblock (4% whey, 15% goat/sheep serum, 0.5% tween 20, 80.5% 1X PBS) for 1hr at room temperature (RT). Plasma samples or monoclonal antibodies were serially diluted in duplicate wells. The corresponding Rh-bNAbs (0.1μg/mL) were included in each plate for standard curve generation. After incubating the plasma at RT for 1hr, the plate was washed with wash buffer, followed by probing the bound antibodies with an anti-monkey IgG horseradish peroxidase (HRP)-conjugated secondary antibody at 1:4000 dilution (Southern Biotech, Birmingham, AL) for 1hr at RT. After washing the plate, a KPL SureBlue substrate (SeraCare Life Sciences, Milford MA) was added for antibody detection and the optical density at 450nm was read using Synergy LX Multimode reader (BioTek, Winooski, VT). The plasma bNAb concentrations were calculated from standard curves using the Gen5 Software (v. 3.12.08). The PK parameters were calculated by performing a non-compartmental analysis using PKanalix program (version 2021 R1. Antony, France: Lixoft SAS 2021).

## Antibody fragmentation

300μg of infused Rh bNAbs 3BNC117, PGT151 or PGT1400 were digested with IdeZ protease (Promega, Madison, WI) for 24hr at 37°C to yield F(ab')2 and Fc fragments. Digested fragments were separated by size exclusion chromatography on AKTA PURE chromatography system (Cytiva Life Sciences) using Superdex 200 increase 10/300 columns (Cytiva Life Sciences). Briefly, 300μg digested antibody was injected into the column and eluted in PBS at a flow rate of 0.5 ml/min at 10°C into 0.5 ml fractions. Eluted fractions were concentrated using Amicon Ultra Centrifugal filters (Millipore Sigma, Burlington, MA).

## Anti-drug antibody (ADA) measurement

The plasma levels of the anti-drug antibodies (ADA) against the administered bNAbs were detected using ELISA as described above. For this assay, 384-well polystyrene plates (Corning Incorporated) were coated overnight with 45ng/well of either total, or purified F(ab')2 or Fc regions of Rh bNAbs 3BNC117, PGT151 or PGDM1400 (UMassBiologics, Boston, MA) and the development of ADA in plasma samples were detected using a goat anti-human lambda IgG HRP-conjugated secondary antibody at 1:4000 dilution (Southern Biotech, Birmingham, AL). The plasma ADA levels were reported as endpoint titer, calculated by determining the lowest dilution that had optical density (OD) greater than five-fold of that in the background wells.

## CH505 SOSIP-specific IgG response

CH505 T/F SOSIP-specific IgG response in the plasma was measured using ELISA as described above. HIV-specific monoclonal antibody 2G12 (produced in-house) coating of the ELISA plate was used as a mechanism to capture HEK 293F cell-generated CH505 T/F SOSIP antigen. The SOSIP-specific IgG response in the serially diluted RM plasma was measured using an anti-monkey IgG horseradish peroxidase (HRP)-conjugated secondary antibody at 1:4000 dilution (Southern Biotech, Birmingham, AL) and concentration of the CH505-SOSIP-

specific IgG response was estimated using a standard curve of CH505 T/F SOSIP IgG detection by Rh HIV-specific monoclonal antibody, b12R1 (in-house).

## Detection of mucosal bNAb levels

To extract mucosal antibody secretions, Weck sponges were transferred to SPIN-X insert with no filter membrane, and saliva or rectal secretions were extracted using prechilled buffer solution (1X PBS, 1X solution of Protease Inhibitor Cocktail and 0.25% Bovine Serum Albumin). Extracted mucosal secretions were concentrated using Amicon Ultra-0.5 Centrifugal Filter Unit (Millipore Sigma, St. Louis, MO). To detect bNAbs in mucosal secretions an anti-idiotypic antibody ELISA as described above was performed. The bNAb concentrations in mucosal secretions were measured using a semi-quantitative ultrasensitive single molecule counting (SMC) platform (Millipore Sigma, St. Louis, MO) using manufacturer's instructions. In absence of the total IgG levels in saliva and rectal swab, this assay was semi-quantitative, where Rh-bNAbs measured in each compartment and time point were assigned a detection strength, based on the limit of detection of the assay and estimated bNAb concentrations.

## Neutralization assay

Neutralization of SHIVs by single or combination of bNAbs and plasma antibodies in TZM-bl cells was measured as previously described [45]. Briefly, RM plasma and/or bNAbs were incubated with SHIV pseudoviruses or infectious molecular clones for 1 h at 37°C and 5% CO2. TZM-bl cells were then added and incubated at 37°C and 5% CO2 for 48 h. Commercially available luciferase reagent (Bright-Glo; Promega, Madison, WI) was added and luminescence was measured using the Synergy LX multilabel plate reader (Agilent, Santa Clara, CA). The 50% and 80% inhibitory concentrations ($IC_{50}$ and $IC_{80}$) of the bNAbs and the 50% and 80% inhibitory dilutions ($ID_{50}$ and $ID_{80}$) of the RM plasma were reported.

## Antibody-dependent cellular cytotoxicity (ADCC)

BNAbs and RM plasma samples were tested using the Luciferase-based (Luc) ADCC assay against the HIV-1 CH505 T/F infectious molecular clone (IMC) adapting previous methodology to derive HIV-1 IMC-infected target cells [46]. This is an ecto-IMC generated using the HIV-1 NL4-3 backbone with the insertion of subtype C HIV-1 CH505 T/F envelope and the Luciferase reporter genes [47] (kindly provided by C. Ochsenbauer, University of Alabama). PBMCs from healthy donors were utilized as source of effector cells. RM plasma was tested for ADCC activity starting at 1:100 dilution using 6 four-fold dilutions of the samples. BNAbs were tested starting at 50μg/mL, using 6 five-fold dilutions. The effector-to-target cell ratio was 30:1. The final readout was the luminescence intensity generated by the presence of residual intact target cells that have not been lysed by the effector population in the presence of ADCC-mediating antibodies. The percentage of killing was calculated using the formula % killing = ((RLU of target and effector well—RLU of test well)/RLU of target and effector well) x 100. In this analysis, the RLU of the target plus effector wells represented spontaneous lysis in the absence of any source of Ab. The analysis of the results was conducted after subtracting the background detected with the pre-infusion samples or the influenza-specific rhesusized CH65-IgG1 (anti-HA) antibody, used as negative controls, from the activity observed with the sera and monoclonal antibodies, respectively. After background subtraction, results were considered positive if the percent specific killing was above 10%. The results were reported as ADCC plasma and monoclonal antibody titers, the latter as μg/mL, calculated based on the interpolation of the plasma antibody titration with the 10% cut-off.

## Antibody-dependent cell phagocytosis (ADCP)

Antibody-dependent cellular phagocytosis (ADCP) assay was performed as previously described [48]. Briefly, CH505 T/F antigen was covalently bound to fluorescent NeutrAvidin beads (Invitrogen, Waltham, MA). Rh PGT151 IgG, PGDM1400 IgG, and 3BNC117 IgG (UMass Biologics, Boston, MA) were diluted to a final concentration of 50 µg/mL and incubated for 2 hours with antigen-conjugated florescent beads to form immune complexes. Immune complexes were then subjected to spinoculation at 1200 g in the presence of a human-derived monocyte cell line, THP-1 cells (ATCC), for 1 h at 4˚C. Following spinoculation, antigens and cells were incubated at 37˚C for phagocytosis. After incubation, THP-1 cells were fixed with 4% paraformaldehyde (Sigma, St. Louis, MO) and cell fluorescence was assessed using flow cytometry (LSR Fortessa; BD). The CD4 binding site-specific bNAb VRC01 was used as a positive control and the influenza-specific MAb CH65 as a negative control. A no-antibody control made of 0.1% phosphate-buffered saline supplemented with 0.1% bovine serum albumin (BSA) was used to determine the background phagocytosis activity. Phagocytosis scores were calculated by multiplying the mean fluorescence intensity (MFI) and frequency of bead-positive cells and dividing by the MFI and frequency of bead-positive cells in the antibody-negative control (PBS). All bNAbs were tested in two independent assays, and the average phagocytosis scores from these two independent assays were reported. The limit of detection of the assay was calculated as 3 times the ADCP score of negative control antibody, CH65.

## $PT_{80}$ calculation and statistical analysis

$PT_{80}$ for the triple bNAb combination against multiple HIV and SHIV variants were computed using previously published Bliss-Hill modeling [49] that utilizes the plasma bNAb concentrations (**Fig 2A–2C**) and Los Almos National Laboratory (LANL) database-reported [50] or experimental $IC_{80}$ of the individual bNAbs against the target viral strains. The correlation between $PT_{80}$ values of the combination bNAb and plasma $ID_{80}$ levels was assessed using the Kendall rank correlation test. All statistical analyses were conducted on the R statistical computing platform [https://www.r-project.org/].

## Supporting information

**S1 Fig. Specificity of the ADA responses against the F(ab)2 and Fc regions of the three infused antibodies.**
(TIF)

**S2 Fig. $PT_{80}$ of the triple bNAb combination against cross-clade SHIV variants.** $PT_{80}$ values of Rh bNAb combination against (**A**) SHIV.A.BG505.332N.375Y.dCT (**B**) SHIV.A.BG505.375Y.dCT (**C**) SHIV.B.WITO.4160.33.375W.dCT (**D**) SHIV.B.B41.375H.dCT (**E**) SHIV.C.Ce.1086c.375S.dCT (**F**) SHIV.C.CH848.375S.dCT (**G**) SHIV.C.CH505.375H.dCT (**H**) SHIV.C.ZM233.PB6.375Y.dCT and (**I**) SHIV.D.191859.375M.dCT. Dashed line represents $PT_{80}>200$ of the antibody combination against the SHIV variants in rhesus macaques.
(TIF)

## Acknowledgments

We thank Drs. David Montefiori (Duke University), Celia LeBranche (Duke University), Kevin Saunders (Duke University) and Katharine Barr (University of Pennsylvania) for providing us with plasmids for producing SHIVs. The following reagent was obtained through the

NIH HIV Reagent Program, Division of AIDS, NIAID, NIH: CEM.NKR CCR5+ Cells, ARP-4376, contributed by Dr. Alexandra Trkola. We thank Weill Cornell Medicine Molecular Biophysics Core and Dr. Radda Rusinova for their expertise in Size Exclusion Chromatography.

## Author Contributions

**Conceptualization:** Sedem Dankwa, Christina Kosman, Maria Dennis, Sallie R. Permar, Genevieve G. Fouda, Ria Goswami.

**Data curation:** Christian Binuya, Carolyn Weinbaum, Ria Goswami.

**Formal analysis:** Sedem Dankwa, Christina Kosman, Maria Dennis, Elena E. Giorgi, Kenneth Vuong, Ioanna Pahountis, Ashley Garza, Christian Binuya, Janice McCarthy, Bryan T. Mayer, Julia T. Ngo, Chiamaka A. Enemuo, Diane G. Carnathan, Sherry Stanfield-Oakley, Stella J. Berendam, Ria Goswami.

**Funding acquisition:** Cliburn Chan, Guido Ferrari, Guido Silvestri, Rama R. Amara, Ann Chahroudi, Sallie R. Permar, Genevieve G. Fouda.

**Investigation:** Sedem Dankwa, Christina Kosman, Maria Dennis, Kenneth Vuong, Ioanna Pahountis, Ashley Garza, Christian Binuya, Janice McCarthy, Julia T. Ngo, Chiamaka A. Enemuo, Diane G. Carnathan, Sherry Stanfield-Oakley, Stella J. Berendam.

**Methodology:** Sedem Dankwa, Christina Kosman, Maria Dennis, Kenneth Vuong, Ioanna Pahountis, Ashley Garza, Christian Binuya, Janice McCarthy, Bryan T. Mayer, Julia T. Ngo, Chiamaka A. Enemuo, Diane G. Carnathan, Sherry Stanfield-Oakley, Stella J. Berendam.

**Project administration:** Carolyn Weinbaum, Ria Goswami.

**Resources:** Kathleen Engelman, Diogo M. Magnani, Guido Ferrari, Guido Silvestri, Rama R. Amara, Ann Chahroudi, Sallie R. Permar, Genevieve G. Fouda.

**Software:** Elena E. Giorgi, Janice McCarthy, Bryan T. Mayer.

**Supervision:** Cliburn Chan, Guido Ferrari, Guido Silvestri, Rama R. Amara, Ann Chahroudi, Sallie R. Permar, Genevieve G. Fouda, Ria Goswami.

**Validation:** Elena E. Giorgi, Carolyn Weinbaum, Ria Goswami.

**Visualization:** Ria Goswami.

**Writing – original draft:** Sedem Dankwa, Christina Kosman, Maria Dennis, Ria Goswami.

**Writing – review & editing:** Sedem Dankwa, Christina Kosman, Maria Dennis, Elena E. Giorgi, Kenneth Vuong, Ioanna Pahountis, Ashley Garza, Janice McCarthy, Bryan T. Mayer, Julia T. Ngo, Chiamaka A. Enemuo, Diane G. Carnathan, Sherry Stanfield-Oakley, Stella J. Berendam, Carolyn Weinbaum, Kathleen Engelman, Diogo M. Magnani, Cliburn Chan, Guido Ferrari, Guido Silvestri, Rama R. Amara, Ann Chahroudi, Sallie R. Permar, Genevieve G. Fouda, Ria Goswami.

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
