## [Decision Letter · Decision Letter 0]

6 Mar 2024

PONE-D-23-34528A novel HIV triple broadly neutralizing antibody (bNAb) combination-based passive immunization of infant rhesus macaques achieves durable protective plasma neutralization levels and mediates anti-viral effector functions.PLOS ONE

Dear Dr. Goswami,

Thank you for submitting your manuscript to PLOS ONE. After careful consideration, we feel that it has merit but does not fully meet PLOS ONE’s publication criteria as it currently stands. Therefore, we invite you to submit a revised version of the manuscript that addresses the points raised during the review process.

We look forward to receiving your revised manuscript.

Kind regards,

Roland Zahn

Academic Editor

PLOS ONE

Journal Requirements:

"This work was supported by NIH P01 5P01AI131276 (S.R.P, G.G.F, A.C, G.S, R.R.A. C.C), NIH 5UM1AI164566 (A.C, S.R.P, G.G.F), NIH 5P01AI117915 (S.R.P). and NIH’s Office of the Director, Office of Research Infrastructure Programs, P51OD011132. The ADCC assay was performed at Duke Human Vaccine Institute (Durham, NC) as part of the Primate AIDS Vaccine Evaluation Group (PAVEG) contract HSN272201800004C. The rhesusization, expression, and purification of antibodies were partially funded by NIH P40 OD028116 (D.M.M., K.E.). The funders had no role in study design, data collection and interpretation, or the decision to submit the work for publication."

"This work was supported by NIH P01 5P01AI131276 (S.R.P, G.G.F, A.C, G.S, R.R.A. C.C), NIH 5UM1AI164566 (A.C, S.R.P, G.G.F), NIH 5P01AI117915 (S.R.P). and NIH’s Office of the Director, Office of Research Infrastructure Programs, P51OD011132. The ADCC assay was performed at Duke Human Vaccine Institute (Durham, NC) as part of the Primate AIDS Vaccine Evaluation Group (PAVEG) contract HSN272201800004C. The rhesusization, expression, and purification of antibodies were partially funded by NIH P40 OD028116 (D.M.M., K.E.).

"Please include your amended statements within your cover letter; we will change the online submission form on your behalf.

"This work was supported by NIH P01 5P01AI131276 (S.R.P, G.G.F, A.C, G.S, R.R.A. C.C), NIH 5UM1AI164566 (A.C, S.R.P, G.G.F), NIH 5P01AI117915 (S.R.P). and NIH’s Office of the Director, Office of Research Infrastructure Programs, P51OD011132. The ADCC assay was performed at Duke Human Vaccine Institute (Durham, NC) as part of the Primate AIDS Vaccine Evaluation Group (PAVEG) contract HSN272201800004C. The rhesusization, expression, and purification of antibodies were partially funded by NIH P40 OD028116 (D.M.M., K.E.). "

"S.R.P serves as a consultant for Merck, Moderna, Pfizer, Hoopika, Dynavax, and GSK on their CMV vaccine programs, and has led sponsored programs with Merck and Moderna on CMV vaccines. Other authors have no conflict of interest to disclose."

Additional Editor Comments:

Please revise figure 2: the ADA induction is graphed in a way that is not reflecting the actual data. ADA presence is only determined experimentally to appear on day 35 while the graph indicates a linear rise from day 3 to day 35, which is likely not correct. 

Reviewers' comments:

Reviewer's Responses to Questions

**Comments to the Author**

1. Is the manuscript technically sound, and do the data support the conclusions?

Reviewer #1: No

Reviewer #2: Partly

2. Has the statistical analysis been performed appropriately and rigorously? 

Reviewer #1: Yes

Reviewer #2: Yes

3. Have the authors made all data underlying the findings in their manuscript fully available?

Reviewer #1: No

Reviewer #2: Yes

4. Is the manuscript presented in an intelligible fashion and written in standard English?

Reviewer #1: Yes

Reviewer #2: Yes

5. Review Comments to the Author

Reviewer #1: Dankwa et al. assessed the safety, pharmacokinetics, ex vivo neutralizing capacity, and functionality of a trivalent rhesuzised HIV bNAb combination that was passively transferred to infant rhesus monkeys. Available data suggest that the treatment is well tolerated, however, the study consists of only three animals. The half-life of bNAbs was relatively short, and was substantially shortened in one animal, coinciding with the generation of ADA. Analysis of serum samples showed intact binding, neutralizing, ADCC and ADCP function of the transferred bNAb cocktail. bNAbs were also detected in mucosal samples, which may be relevant for protection from vertical transmission. Preventive efficacy of the treatment against a panel of HIV variants was calculated using the PK-data-derived PT80 biomarker, which correlated with the experimentally derived ID80 titers.

The manuscript is well written and clear. The data provide insight into the PK and tissue penetration of bNAbs in the infant NHP model, however, it is not convincingly justified how the PT80 biomarker can give any indication of the PE to be expected in bNAb treated infants.

Comments:

1. The rapid generation of ADA in 2 out of 3 NHP is a concern. Is this related to the young age of the animals and is this frequency also expected in human infants? It would be helpful if this aspect was discussed in greater detail.

2. Please clarify in the manuscript how data in figure 4B were generated for the present experiment. Are these data based on average serum Ab concentrations across all three RMs, or only for one representative animal?

3. The authors predict an PE of over 90% based on the PT80 biomarker. This prediction is in my view incorrect since it assumes that serum PT80 similarly predicts efficacy against adult (sexual) transmission, as observed in AMP, and infant horizontal transmission. Either the authors need to convincingly justify why PT80 should similarly predict PE for both routes of transmission, or the assumption ‘that universal infant prophylaxis early-after birth, in areas of HIV seroprevalence could provide protection from perinatal and breastmilk HIV transmission’ should be removed, and, throughout the whole manuscript, the interpretation of the PT80 biomarker data should not extrapolate to efficacy in infants.

Reviewer #2: Dankwa et al. describe an important study investigating the pharmacokinetics and potency of an optimized combination of three broadly neutralizing antibodies (bnAbs) in infant rhesus macaques. The study is well designed, experiments mostly technically sound, and the manuscript is well written. The manuscript can benefit from addressing the following points:

1. Fig. 1 should be labeled to specifiy that SHIV.C.CH505.375H.dCT was used for neutralization assay.

2. The authors should repeat all neutralization assay using full length, WT CH505 Env (Fig. 3B) in addition to the data they present for the variant with the cytoplasmic tail deleted.

3. It would be helpful to provide more data on the ADCC assay, including raw data, and calculations.

4. It would be helpful to identify the epitope(s) of the anti-drug antibodies developed in the rhesuses macaques.

6. PLOS authors have the option to publish the peer review history of their article (what does this mean?). If published, this will include your full peer review and any attached files.

Reviewer #1: **Yes: **Frank Wegmann

Reviewer #2: No

---

## [Author Response · Author response to Decision Letter 0]

2 Sep 2024

Journal Requirements:

Response: Thank you for this suggestion. We have now formatted our manuscript based on PLOS One’s style requirement and named the files as suggested.

"This work was supported by NIH P01 5P01AI131276 (S.R.P, G.G.F, A.C, G.S, R.R.A. C.C), NIH 5UM1AI164566 (A.C, S.R.P, G.G.F), NIH 5P01AI117915 (S.R.P). and NIH’s Office of the Director, Office of Research Infrastructure Programs, P51OD011132. The ADCC assay was performed at Duke Human Vaccine Institute (Durham, NC) as part of the Primate AIDS Vaccine Evaluation Group (PAVEG) contract HSN272201800004C. The rhesusization, expression, and purification of antibodies were partially funded by NIH P40 OD028116 (D.M.M., K.E.). The funders had no role in study design, data collection, and interpretation, or the decision to submit the work for publication."

"This work was supported by NIH P01 5P01AI131276 (S.R.P, G.G.F, A.C, G.S, R.R.A. C.C), NIH 5UM1AI164566 (A.C, S.R.P, G.G.F), NIH 5P01AI117915 (S.R.P). and NIH’s Office of the Director, Office of Research Infrastructure Programs, P51OD011132. The ADCC assay was performed at Duke Human Vaccine Institute (Durham, NC) as part of the Primate AIDS Vaccine Evaluation Group (PAVEG) contract HSN272201800004C. The rhesusization, expression, and purification of antibodies were partially funded by NIH P40 OD028116 (D.M.M., K.E.).

"Please include your amended statements within your cover letter; we will change the online submission form on your behalf.

Response: Thank you for this suggestion. We have now removed any funding-related information from the manuscript's text and the Acknowledgement section. The online funding statement will be updated to the following. 

"This work was supported by NIH P01 5P01AI131276 (S.R.P, G.G.F, A.C, G.S, R.R.A. C.C), NIH 5UM1AI164566 (A.C, S.R.P, G.G.F), NIH 5P01AI117915 (S.R.P). and NIH’s Office of the Director, Office of Research Infrastructure Programs, P51OD011132. The ADCC assay was performed at Duke Human Vaccine Institute (Durham, NC) as part of the Primate AIDS Vaccine Evaluation Group (PAVEG) contract HSN272201800004C. The rhesusization, expression, and purification of antibodies were partially funded by NIH P40 OD028116 (D.M.M., K.E.). The funders had no role in study design, data collection and analysis, decision to publish, or preparation of the manuscript."

We have also included this updated version in the cover letter.

"This work was supported by NIH P01 5P01AI131276 (S.R.P, G.G.F, A.C, G.S, R.R.A. C.C), NIH 5UM1AI164566 (A.C, S.R.P, G.G.F), NIH 5P01AI117915 (S.R.P). and NIH’s Office of the Director, Office of Research Infrastructure Programs, P51OD011132. The ADCC assay was performed at Duke Human Vaccine Institute (Durham, NC) as part of the Primate AIDS Vaccine Evaluation Group (PAVEG) contract HSN272201800004C. The rhesusization, expression, and purification of antibodies were partially funded by NIH P40 OD028116 (D.M.M., K.E.). "

Response: Thank you for this suggestion. We have updated the financial disclosure as suggested (find below) and have included it in the cover letter.

"This work was supported by NIH P01 5P01AI131276 (S.R.P, G.G.F, A.C, G.S, R.R.A. C.C), NIH 5UM1AI164566 (A.C, S.R.P, G.G.F), NIH 5P01AI117915 (S.R.P). and NIH’s Office of the Director, Office of Research Infrastructure Programs, P51OD011132. The ADCC assay was performed at Duke Human Vaccine Institute (Durham, NC) as part of the Primate AIDS Vaccine Evaluation Group (PAVEG) contract HSN272201800004C. The rhesusization, expression, and purification of antibodies were partially funded by NIH P40 OD028116 (D.M.M., K.E.). The funders had no role in study design, data collection and analysis, decision to publish, or preparation of the manuscript."

"S.R.P serves as a consultant for Merck, Moderna, Pfizer, Hoopika, Dynavax, and GSK on their CMV vaccine programs, and has led sponsored programs with Merck and Moderna on CMV vaccines. Other authors have no conflict of interest to disclose."

Response: Thanks for bringing this up. We have now updated our Competing interest section to the following. 

“S.R.P serves as a consultant for Merck, Moderna, Pfizer, Hoopika, Dynavax, and GSK on their CMV vaccine programs, and has led sponsored programs with Merck and Moderna on CMV vaccines. Other authors have no conflict of interest to disclose. This does not alter our adherence to PLOS ONE policies on sharing data and materials.”

We have also included this in our cover letter and in the manuscript text in lines 673-676.

Response: Thank you for the suggestion. All data has been uploaded on Github, accessible from github.com/Riagoswami1/Triple_bNAb_PK_infant_NHP. Data access will be made public, upon provisional acceptance of the manuscript. This information has now been updated in the manuscript text in line 649-650.

Additional Editor Comments:

Please revise figure 2: the ADA induction is graphed in a way that is not reflecting the actual data. ADA presence is only determined experimentally to appear on day 35 while the graph indicates a linear rise from day 3 to day 35, which is likely not correct. 

Response: We thank the Editor for this suggestion. We have now changed Figure 2 per Editor’s suggestion. Figure 2A-C now indicate the levels of the 3 antibodies in plasma and Figure 2D-F demonstrate ADA levels against the 3 antibodies. We have also made the necessary changes in the figure legend (lines 187-192).

Reviewers' comments:

Reviewer's Responses to Questions

Comments to the Author

1. Is the manuscript technically sound, and do the data support the conclusions?

Reviewer #1: No

Reviewer #2: Partly

2. Has the statistical analysis been performed appropriately and rigorously? 

Reviewer #1: Yes

Reviewer #2: Yes

3. Have the authors made all data underlying the findings in their manuscript fully available?

Reviewer #1: No

Reviewer #2: Yes

4. Is the manuscript presented in an intelligible fashion and written in standard English?

Reviewer #1: Yes

Reviewer #2: Yes

5. Review Comments to the Author

Reviewer #1: Dankwa et al. assessed the safety, pharmacokinetics, ex vivo neutralizing capacity, and functionality of a trivalent rhesuzised HIV bNAb combination that was passively transferred to infant rhesus monkeys. Available data suggest that the treatment is well tolerated, however, the study consists of only three animals. The half-life of bNAbs was relatively short, and was substantially shortened in one animal, coinciding with the generation of ADA. Analysis of serum samples showed intact binding, neutralizing, ADCC and ADCP function of the transferred bNAb cocktail. bNAbs were also detected in mucosal samples, which may be relevant for protection from vertical transmission. Preventive efficacy of the treatment against a panel of HIV variants was calculated using the PK-data-derived PT80 biomarker, which correlated with the experimentally derived ID80 titers.

The manuscript is well written and clear. The data provide insight into the PK and tissue penetration of bNAbs in the infant NHP model, however, it is not convincingly justified how the PT80 biomarker can give any indication of the PE to be expected in bNAb treated infants.

Response. We agree with the reviewer's concern that the PT80 biomarker has not been established to predict prevention from perinatal and breast milk transmission. We have modified the manuscript text to highlight that future studies to investigate whether a PT80 biomarker threshold of >200 against perinatally and breast milk-exposed HIV variants is sufficient to provide ≥90% preventative efficacy against vertical HIV transmission will be crucial (Line 441-447). We have removed the claim that in perinatal or breast milk transmission PT80>200 is equivalent to PE≥90%. We have modified the manuscript text and updated Figure 4B and 4D and associated legends to reflect the change.

Comments:

1. The rapid generation of ADA in 2 out of 3 NHP is a concern. Is this related to the young age of the animals and is this frequency also expected in human infants? It would be helpful if this aspect was discussed in greater detail.

Response. We thank the reviewers for these questions. We have now included a paragraph (Lines 390-409) to discuss the potential contribution of age, immunization regimen, dose, route and frequency of administration, and host-specificity on the development of ADA responses against infused bNAbs. 

2. Please clarify in the manuscript how data in figure 4B were generated for the present experiment. Are these data based on average serum Ab concentrations across all three RMs, or only for one representative animal?

Response. We apologize for the lack of clarity in our manuscript methods. The data shown in Figure 4B are based on median PT80 values of the combination regimen for each SHIV variant. We have now clarified this in Figure 4B legend.

3. The authors predict an PE of over 90% based on the PT80 biomarker. This prediction is in my view incorrect since it assumes that serum PT80 similarly predicts efficacy against adult (sexual) transmission, as observed in AMP, and infant horizontal transmission. Either the authors need to convincingly justify why PT80 should similarly predict PE for both routes of transmission, or the assumption ‘that universal infant prophylaxis early-after birth, in areas of HIV seroprevalence could provide protection from perinatal and breastmilk HIV transmission’ should be removed, and, throughout the whole manuscript, the interpretation of the PT80 biomarker data should not extrapolate to efficacy in infants.

Response. We agree with the reviewer's concern that the PT80 biomarker has not been established to predict prevention from perinatal and breast milk transmission. We have modified the manuscript text to highlight that future studies to investigate whether a PT80 biomarker threshold of >200 against perinatally and breast milk-exposed HIV variants is sufficient to provide ≥90% preventative efficacy against vertical HIV transmission will be crucial (Line 441-447). We have removed the claim that in perinatal or breast milk transmission PT80>200 is equivalent to PE≥90%. We have modified the manuscript text and updated Figure 4B and 4D and associated legends to reflect the change.

Reviewer #2: Dankwa et al. describe an important study investigating the pharmacokinetics and potency of an optimized combination of three broadly neutralizing antibodies (bnAbs) in infant rhesus macaques. The study is well designed, experiments mostly technically sound, and the manuscript is well written. The manuscript can benefit from addressing the following points:

1. Fig. 1 should be labeled to specifiy that SHIV.C.CH505.375H.dCT was used for neutralization assay.

Response. We thank the reviewer for this suggestion. Figure 1A has now been re-labeled.

2. The authors should repeat all neutralization assay using full length, WT CH505 Env (Fig. 3B) in addition to the data they present for the variant with the cytoplasmic tail deleted.

Response. We thank the reviewer for this suggestion. In a previous manuscript from our group (PMID: 33177194), we demonstrated that the neutralization potency of the 3 infused bNAbs against WT CH505 infectious molecular clone was well correlated with that of the SHIV.CH505.375H.dCT variant (Figure 1 of PMID 33177194). Since SHIV.C.CH505.375H.dCT is 

---

## [Editor Report · Decision Letter 1]

7 Oct 2024

A novel HIV triple broadly neutralizing antibody (bNAb) combination-based passive immunization of infant rhesus macaques achieves durable protective plasma neutralization levels and mediates anti-viral effector functions.

PONE-D-23-34528R1

Dear Dr. Goswami,

We’re pleased to inform you that your manuscript has been judged scientifically suitable for publication and will be formally accepted for publication once it meets all outstanding technical requirements.

Kind regards,

Roland Zahn

Academic Editor

PLOS ONE
---

## [Editor Report · Acceptance letter]

30 Oct 2024

PONE-D-23-34528R1 

PLOS ONE

Dear Dr. Goswami, 

I'm pleased to inform you that your manuscript has been deemed suitable for publication in PLOS ONE. Congratulations! Your manuscript is now being handed over to our production team.

Kind regards, 

on behalf of

Dr. Roland Zahn 

Academic Editor

PLOS ONE